# Plasma nanoDSF Denaturation Profile at Baseline Is Predictive of Glioblastoma EGFR Status

**DOI:** 10.3390/cancers15030760

**Published:** 2023-01-26

**Authors:** Rémi Eyraud, Stéphane Ayache, Philipp O. Tsvetkov, Shanmugha Sri Kalidindi, Viktoriia E. Baksheeva, Sébastien Boissonneau, Carine Jiguet-Jiglaire, Romain Appay, Isabelle Nanni-Metellus, Olivier Chinot, François Devred, Emeline Tabouret

**Affiliations:** 1Laboratoire Hubert Curien UMR 5516, UJM-Saint-Etienne, University Lyon, CNRS, 42000 Saint Etienne, France; 2LIS, Aix Marseille Univ, CNRS, 13288 Marseille, France; 3Inst Neurophysiopathol, INP, Faculté des Sciences Médicales et Paramédicales, Aix Marseille Univ, CNRS, 13005 Marseille, France; 4Plateforme Interactome Timone, PINT, Faculté des Sciences Médicales et Paramédicales, Aix Marseille Univ, 13005 Marseille, France; 5Department of Neuro-Surgery, Timone Hospital, APHM, 13005 Marseille, France; 6Department of Anatomopathology, Timone Hospital, APHM, 13005 Marseille, France; 7Department of Molecular Oncology, APHM, 13016 Marseille, France; 8Service de Neurooncologie, CHU Timone, APHM, 13005 Marseille, France

**Keywords:** cancer, glioblastoma, differential scanning fluorimetry, machine learning, plasma profiling, biomarker, neurophysiopathology, neuro-oncology

## Abstract

**Simple Summary:**

The toolkit for diagnosing the most aggressive primary brain tumor glioblastoma (GBM) is very limited. We recently demonstrated that plasma denaturation profiles (PDPs) of GBM patients and healthy controls obtained with nanoDSF can be automatically classified using artificial intelligence (AI) algorithms. Since PDPs have been shown to be useful for subtype differentiation for lung cancer, we decided to investigate whether nanoDSF-derived PDPs could also be used to discriminate EGFR alterations in GBM, which is important for determining therapy strategies. We found that AI is able to discriminate EGFR alterations in GBM with an 81.5% accuracy. Thus, we have demonstrated that the use of plasma denaturation profiles could answer the unsatisfied neuro-oncology need for a predictive diagnostic biomarker, which could complete MRI and clinical data, allowing for a rapid orientation of patients for a definitive pathological diagnosis and treatment.

**Abstract:**

Glioblastoma (GBM) is the most frequent and aggressive primary brain tumor in adults. Recently, we demonstrated that plasma denaturation profiles of glioblastoma patients obtained using Differential Scanning Fluorimetry can be automatically distinguished from healthy controls with the help of Artificial Intelligence (AI). Here, we used a set of machine-learning algorithms to automatically classify plasma denaturation profiles of glioblastoma patients according to their EGFR status. We found that Adaboost AI is able to discriminate *EGFR* alterations in GBM with an 81.5% accuracy. Our study shows that the use of these plasma denaturation profiles could answer the unmet neuro-oncology need for diagnostic predictive biomarker in combination with brain MRI and clinical data, in order to allow for a rapid orientation of patients for a definitive pathological diagnosis and then treatment. We complete this study by showing that discriminating another mutation, MGMT, seems harder, and that post-surgery monitoring using our approach is not conclusive in the 48 h that follow the surgery.

## 1. Introduction

*IDH* wild-type *(IDHwt)* glioblastoma (GBM) is the most frequent and aggressive primary brain tumor. Its diagnosis is based on histological analyses of tumor samples showing the presence of angiogenesis and/or necrosis [1]. Currently, the first-line treatment combines a surgical resection, if accessible, followed by radio-chemotherapy [2]. Despite standard-of-care administration, relapse is inevitable in a median delay of 7 to 10 months. When tumor resection is possible, precious molecular information can be obtained from resected tissue that can guide either patient diagnosis or prognosis [3]. For instance, the presence of specific molecular alterations, such as Epithelial Growth Factor Receptor gene (*EGFR*) amplification or mutation, signifies a grade IV diagnosis, highlighting the crucial role of glioblastoma molecular characterization [4]. Additionally, patients with a methylated Methylguanine methyltransferase (*MGMT)* promoter in tumor samples present a better prognosis and higher temozolomide sensitivity due to MGMT enzyme inactivation [5]. However, in 10 to 20% of patients, glioblastoma is not accessible for surgical resection due to the tumor’s deep location or patient morbidities, thus preventing the histological and molecular characterization of the tumor [5,6]. This can lead to a significant delay or even failure of diagnosis, which has a great negative impact on patient management and survival. Finding a way to access this information without a biopsy is an unmet need [2,6].

Over the past 15 years, it has been shown that Plasma Denaturation Profiles (PDPs) obtained by Differential Scanning Calorimetry (DSC) [7] could be used to obtain disease-specific signatures for a wide variety of pathologies [8,9], including glioblastoma [10]. Moreover, it was demonstrated that PDPs obtained by DSC could even be used for subtype differentiation and survival prediction [10,11]. Recently, we have demonstrated that PDPs obtained using another biophysical method, the Differential Scanning Fluorimetry (nanoDSF), could also be used for diagnostic purposes. Indeed, machine-learning algorithms allowed us to automatically distinguish nanoDSF PDPs of glioma patients from healthy controls with a 92% accuracy [12]. 

In this study, we now apply nanoDSF technology to finer aspects of GBM diagnosis characterization. First, we questioned whether different molecular subtypes can change the nanoDSF signature of GBM. Second, we assessed how glioblastoma resection may impact this signature. 

## 2. Methods

### 2.1. Patient Cohort

We established a local prospective cohort composed of 38 adult (≥18 years) *IDHwt* glioblastoma patients included at initial diagnosis between June 2016 and October 2017 at Timone Hospital (Marseille, France). For these patients, the plasma samples were collected before and 48 h after surgical resection. This cohort was completed by 7 patients hospitalized for the resection of a brain aneurysm without glioblastoma.

The following data were recorded: age, gender, type of surgery, Karnofsky Performance Status (KPS), oncological treatment, clinical symptom, steroid dose, and magnetic resonance imaging (MRI) characteristics.

For plasma collection, peripheral blood was drawn into an EDTA tube. Blood samples were centrifuged within 2 h at 3000× *g* for 10 min at room temperature, and plasma was stored at −80 °C. All samples (plasma and formalin-fixed paraffin-embedded (FFPE) tumor samples) were stored in the Assistance Publique des Hopitaux de Marseille (APHM) Biological Resource Center (BRC) (authorization number CRB BB-0033-00097). Tumor and plasma samples were obtained after written consent according to a protocol approved by the local institutional review board and ethic committee. The present studies were conducted in accordance with the declaration of Helsinki.

### 2.2. EGFR Amplification Determination by Next Generation Sequencing

The nucleic acids were extracted with the Maxwell RSC DNA FFPE kit (Promega, Madison, WI, USA) from formalin-fixed, paraffin-embedded (FFPE) tumor samples. For amplification screening, we used our custom panel Oncomine Solid Tumor and Oncomine Solid Tumor+ (OST/OST+), which includes *EGFR*. For all clinical samples, we performed sequencing with Ion Torrent S5XL (ThermoFisher Scientifics, Waltham, MA, USA) with a sensitivity of 5% and a minimum coverage of 500×. Then, data were analyzed through two complementary pipelines. The first pipeline was developed by ThermoFisher on the IonTorrent Suite + Ion Reporter. The second pipeline, which was developed in our laboratory, uses open-source software such as BWA-MEM for alignment, SAMtools for mpileup, VarScan2 as the variant caller and VEP Ensemble for annotations.

### 2.3. MGMT Promoter Methylation Determination by Pyrosequencing

DNA extraction was performed from 5 slides from FFPE tumoral fragments using the QIAamp DNA kit (Qiagen, Courtaboeuf, France). Only samples containing at least 60% of tumor cells were processed (neuropathologist confirmation). A total of 20 to 200 ng of DNA was treated with sodium bisulfite using the EpiJET Bisulfite Conversion kit and purified according to the specified protocol (Thermo Fischer Scientific, Inc.,Waltham, MA, USA). Bisulfit-modified DNA was amplified using ampliTaq Gold 360 Master mix (Applied Biosystems, Foster City, CA, USA) with a forward primer and a biotinylated reverse primer (Pyromark Q96 CpG MGMT, Qiagen, Courtaboeuf, France). Pyrosequencing was performed using PyroMark-Q48 advanced CpG Reagents and the sequencing primer (Pyromark Q96 CpG MGMT Qiagen) using the Pyromark Q48 Autoprep software on a PyroMark Q48 pyrosequencer (Qiagen, Courtaboeuf, France). Full details for the CpG location and the validation method can be found in the study by Quillien et al. [13].

### 2.4. Sample Analysis by NanoDSF

Plasma denaturation profiles were obtained using the Prometheus NT.Plex instrument (Nanotemper), as previously described [12]. Briefly, plasma samples were loaded to 10 μL capillaries and scanned using Prometheus at 5% of laser power and a 1 °C/min heating rate to obtain PDP in a range of 15 to 95 °C. We carefully mixed preoperation and postoperation patients to avoid any batch effect. Raw data, namely fluorescence at 330 and 350 nm (F330 and F350), scattering at 350 nm (S350) as well as the ratio of fluorescence (F330/F350) were exported into datasets. The first derivatives of these 4 measures were added to the datasets in order to provide their dynamics. 

### 2.5. AI Analyses

In order to investigate the differences between PDPs, we used the previously designed training framework [12] to automatically classify the obtained profiles using Artificial Intelligence (AI). Briefly, we tested several machine-learning (ML) algorithms [14] on the raw data: the classical Logistic Regression (LR), the often well-performing Support Vector Machine (SVM), and two different ensemble methods: Random Forest (RF) and Adaptive Boosting (AdaBoost). These algorithms were evaluated using a leave-one-out approach, where each datum is used once as a test, while the others are used to train the automatic classifiers: the obtained values were thus averages of these repeated experiments. The main metric monitored to evaluate the quality of this process is the accuracy (number of correct classifications by an algorithm divided by the total number of samples). We also keep track of the wrongly classified data and their type.

Usually, working with Machine Learning algorithms starts with a phase that aims at optimizing the algorithms on the given data, for instance by hyperparameters or regularization tuning. However, given the available data, we did not follow any particular approach in this work to enhance the reported results: this would have required us to not use all the available data to compute the reported scores, which is something unimaginable given the number of available data.

Instead, we simply used the hyperparameter values that allowed the best results in our previous article [12]. This implies a potential for improvement, while still validating our proof of concept, as shown in the next Section. 

## 3. Results

### 3.1. Patient Characteristics

Thirty-eight patients were included, with a median age of 65.5 years (range: 42.1–85.4). All patients presented *IDHwt* glioblastoma at diagnosis. The majority of patients benefited from surgical resection followed by concomitant radio-chemotherapy and then adjuvant temozolomide (Table 1).

### 3.2. Molecular Profile

We first tested the ability of our approach to predict molecular profiles from PDPs obtained from pre-surgery patient samples. We trained our AIs on the task of detecting *EGFR* amplification from plasma samples, that is, to distinguish PDPs from 16 negatives (normal *EGFR*) and 11 positive (*EGFR*-amplified) patients (Table 2). Two algorithms were able to predict this molecular alteration from PDPs, one, namely Adaboost, achieving an accuracy higher than 80% (Figure 1). Moreover, this algorithm presented only one false positive, which was particularly important due to the classification consequence of *EGFR* alteration occurrence.

Given the success of this first task, we trained our AIs on the task of detecting *MGMT* promoter methylation in pre-surgery samples, but we were not able to make conclusions on the ability of our approach to distinguish the two groups. Indeed, the apparently good predictive value of the SVM algorithm (76%) merely reflected the unevenness of the two populations (18 negative vs. 7 positive patients) (Table 2).

### 3.3. Comparative Analysis before and after Surgery

To evaluate the impact of surgery on the PDPs obtained by nanoDSF, we compared PDPs of preoperative GBM and postoperative GBM. The comparisons of glioblastoma PDPs before and after surgery were easily distinguishable—they were identified using AI with an optimal accuracy of 82% (RF algorithm, Table 2). However, to make sure that surgical resection and its reactional transitory inflammatory response would not impair PDP determination, we used post-operative non-cancer samples of aneurysm as a control. As depicted in Figure 2, while GBM and aneurysm pre-surgery PDPs were clearly different, both GBM and aneurysm post-surgery PDPs were extremely similar, suggesting that post-surgery PDP modifications could be related to neurosurgery and not a specific disease. It should also be noted that preoperative aneurysm PDP is very similar to the healthy individual one (see Figure 1 of [12]).

## 4. Discussion

In the present study, we confirmed the specificity of pre-surgical PDP and raised the issue of post-surgical evaluation performed too early, after a tumor resection and inflammatory reaction following craniectomy. Moreover, we reported the ability of AI on PDPs to discriminate *EGFR* amplification but not the *MGMT* methylation status of GBM.

The early identification of brain tumors constitutes an unmet medical need [15]. The definitive diagnosis requires an aggressive cerebral biopsy of surgery that could be delayed due to patient comorbidities or a tumor’s deep location [16]. However, definitive brain tumor diagnosis remains an emergency, due to the terrible prognosis of these diseases. Currently, no non-invasive detection methods are available and validated. The use of multimodality MRI has significantly improved brain tumor detection and characterization. However, differential neuro-imaging diagnoses are still impairing the accuracy of brain MRI and do not allow its use alone during the early step of patient management [17]. Regarding circulating biomarkers, the detection of circulating tumor cells or circulating tumor DNA (ctDNA) of glioblastoma is limited by the low circulation of these markers, probably due to the blood–brain barrier impairing their blood release [18]. In parallel, no circulating protein signature was specifically associated with brain tumor development. In this context, the accuracy of our PDP signature opens very promising perspectives, in combination with current methods such as MRI, to accelerate brain tumor diagnosis suspicion. Then, the use of PDPs could help in the medical decision of physicians and neurosurgeons to propose more aggressive definitive diagnostic and therapeutic approaches.

Moreover, the last World Health Organization (WHO) classification of diffuse glioma incorporated very important modifications regarding the role of molecular alterations in diagnosis [19]. Now, *IDHwt* astrocytoma with a Telomerase Reverse Transcriptase Promoter (*pTERT)* mutation, *EGFR* alteration or chromosome 7 gain/chromosome 10 loss are automatically classified as grade IV *IDHwt* glioblastoma, despite the absence of histological necrosis and angiogenesis. These breakdown modifications highlight the critical role of molecular characterization, allowing a very fast integrative diagnosis. In the present study, we were able to predict *EGFR* amplification with an accuracy higher than 80%. Previously, the evaluation of circulating tumor DNA was generating a major hope of helping and accelerating brain tumor diagnosis, allowing a non-invasive determination of the molecular profile of brain tumors. This DNA is released due to various cell secretions and cell-damage mechanisms. This ctDNA is supposed to carry the characteristic mutations of the tumor [20]. However, the use of plasma ctDNA in neuro-oncology involves various difficulties, including a relatively low concentration in brain tumors, restricting its detection capacity in a majority of patients [21]. Then, the PDPs could be an interesting surrogate marker of tumor molecular alterations, again helping the medical decision and accelerating the patient treatment in case of an aggressive molecular profile or delaying a risky therapeutic approach in case of a favorable molecular profile.

The biological rationale to explore and use the PDPs is linked to the modification of the protein plasmatic profile of glioblastoma patients. As with other cancers, primary brain tumors are associated with specific plasmatic modifications consecutive to the tumor development [22]. These circulating alterations are limited and distinct from the other systemic cancers, probably due to the presence of the blood–brain barrier (BBB). However, it is now known that if BBB limits the access of the drugs to the whole tumor, it is partially altered in some tumor regions, as the MRI contrast enhancement can attest [23]. These partial modifications could allow for the modification of the circulating systemic compartment [24]. As an example, previous studies reported a systemic inflammation response in glioblastoma patients with an interesting prognostic value [25,26,27,28,29], and we reported that several plasmatic biomarkers were associated with patient prognosis and treatment response [30,31]. Then, we can hypothesize that our PDPs allow us to encompass a global overview of systemic changes related to glioblastoma and then to identify them from healthy people or distinguish their different subtypes, which are associated with distinct metabolic and oncogenic pathways. In contrast, in postoperative settings, the inflammatory marker release following brain surgery limits our ability to discriminate a glioblastoma versus aneurysm biomarker in plasma, leading to similar denaturation profiles [32].

In this context, our approach presents the advantage of being easy, fast and inexpensive. These PDPs could resolve the unmet neuro-oncology need for a diagnostic predictive biomarker, in combination with brain MRI and clinical data, in order to allow for a rapid orientation of our patients for a definitive pathological diagnosis and then treatment. In the present study, we showed that plasma sampling must occur in pre-surgical time for PDPs analyses, validating the analysis plan for future clinical developments to come.

Despite the fact that our cohort was small and monocentric, we were able to distinguish two molecular subgroups, opening up optimistic perspectives in terms of validation in a larger prospective cohort. In addition, the machine-learning algorithms used here come with theoretical guarantees that ensure better results when the number of data increases, and they are usually run on hundreds, or even thousands, of data: the results on the *EGFR* amplification detection on this small number of PDPs is thus remarkable and promising; their failure on the *MGMT* methylation status can be relativized, since the behavior of the algorithms cannot be predicted when more data are available; in addition, there is the imbalanced number of PDPs of each class (18 negative vs. 7 positive patients), which clearly penalizes machine-learning approaches. In addition, the number of data forbids us from using any approach to adapt the AI to the particular tasks at hand, such as hyperparameters or regularization tuning. This likely explains the behavior of the algorithms that failed. At the same time, it emphasizes the presence of useful information in PDPs: if a non-finetuned Adaboost algorithm is able to learn how to detect EGFR amplification, one can be very hopeful for all algorithms once enough data is available to optimize them on the task. Altogether, the presented results provide a proof of concept that EGFR amplification can be detected using PDPs and AI.

## 5. Conclusions

The baseline plasma denaturation profile (PDP) signature is specific to GBM patients and may predict the molecular status of GBM. Larger prospective studies are ongoing to validate these results.

## Figures and Tables

**Figure 1 cancers-15-00760-f001:**
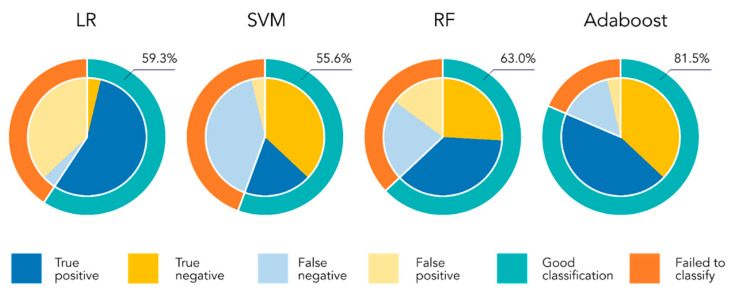
Classification results of the different algorithms on the *EGFR* amplification detection tasks. The external circle of each plot shows the success/failure of each AI. The reported value is the obtained accuracy, that is, the percentage of success. The inside pie presents the 2 classes distribution: the light colors show the proportion of data of each class that are not correctly classified by the algorithm.

**Figure 2 cancers-15-00760-f002:**
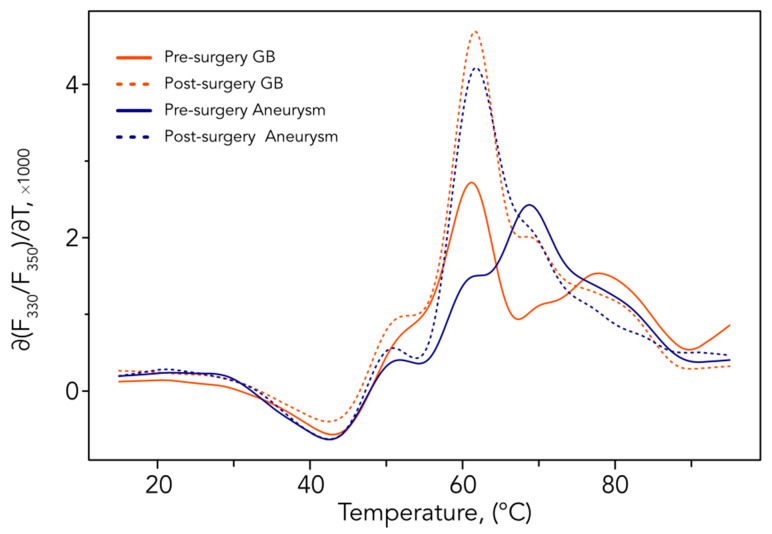
Means of the first derivatives of the ratio F_330_/F_350_. Comparison of aneurysm and GBM patients’ PDPs before and in the 48 h following surgery. All outputs of nanoDSF used for the analysis are shown in Appendix A.

**Table 1 cancers-15-00760-t001:** Patient characteristics.

Factors	Prospective Cohort	
	N = 38	%
Age (median, rage)	65.5 (42.1–85.4)	
Gender (Women/Men)	27/11	71/29
Initial KPS (median, range)	70 (40–100)	
Cognitive symptom	9	24
Steroid doses	40 (10–120)	
Surgery types		
Gross total resection	27	75
Partial resection	9	25
First-line treatment		
Radio-chemotherapy	28	76
Radio-chemotherapy + bevacizumab		
Chemotherapy alone	9	24

**Table 2 cancers-15-00760-t002:** Best obtained accuracy with the different machine learning algorithms.

	N	LR	SVM	RF	Adaboost
*EGFR* amplification (False positive/False negative)	16/11 **	59.3% (10/1)	55.6% (1/11)	63.0% (4/6)	81.5% (1/4)
*MGMT* promoter (False positive/False negative)	18/7 **	48.0% (12/1)	76.0% (1/5)	72.0% (1/6)	60.0% (4/6)
post-/pre-surgery (GBM only) *(False post-surgery/False pre-surgery)*	28/33 *	77.0% (3/11)	80.3% (6/6)	82.0% (4/7)	80.3% (5/7)

* Number of post-surgery/of pre-surgery; ** Number of negatives/of positives.

## Data Availability

The data that were used in this study to obtain the average profiles are available on request from the corresponding author. The medical data are not publicly available due to ethical reasons. The Machine Learning code and the PDPs for the EGFR classification are available at https://github.com/RemiEyraud/EGFR_amplification_detection (accessed on 5 October 2022).

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
