# Peer review of "Plasma nanoDSF Denaturation Profile at Baseline Is Predictive of Glioblastoma EGFR Status"

_cancers, 2023, doi:10.3390/cancers15030760_

Round 1
Reviewer 1 Report
In their manuscript "Plasma nanoDSF denaturation profile at baseline is predictive of glioblastoma EGFR status" Eyraud and collaborators explore the utility of nanoDSF data for molecular characterization of glioblastoma. Although the manuscript contains very interesting data, several issues need to be addressed before the manuscript can be considered for publication.
1) The authors should support their statement in the introduction and discussion section with appropriate references, e.g., the first paragraph of the introduction has only one citation, whereas all included pieces of information are already published and thus should be appropriately referenced. Also, please include details of the citation referenced on page 3, line 102 in the reference section.
2) Methods section 2.1 is missing the details of sample processing, e.g., type of coagulant (for plasma samples), and timeline (how long after collection samples were frozen). Also, based on the descriptions included in other Method sections, it seems that tissue samples were FFPE? If yes, an appropriate description should also be included.
3) Please include the appropriate description of abbreviations (including gene names) when they first appear in the text.
4) The authors should reevaluate their manuscript for consistency in their descriptions. For example, based on the methods section 2.2., The authors analyzed both the amplification and mutations/fusions of EGFR and the corresponding description can be found in Table 2 ("EGFR alterations"). However, in section 3.2, they wrote a statement suggesting that only amplification was targeted by AI algorithms. Does this mean that no mutations/fusions were detected in any of the samples or that authors decided to exclude them (if yes, why?). Or maybe all alterations were included, but the description is incorrect? In other words, what precisely the authors analyzed regarding EGFR?
5) Similarly, the meaning of the word "control", is confusing when reading the manuscript for the first time. The authors first use this word twice when referring to the healthy population used in the previous study, after which they use it again with no detailed description in section 2.4 (page 3), which could be understood as method controls or samples from the healthy population. Only on page 5, the authors finally defined controls as patients with an aneurysm.
6) Why was molecular analysis performed only for 27 samples (for EGFR) or 25 samples (MGMT methylation)? What are the exclusion criteria?
7) In section 2.2 the authors describe two pipelines used to analyze the EGFR alteration data. Are they complementary or used in parallel? If in parallel, what was the authors' approach when conflicting results were obtained using both methods?
8) According to section 2.3, the authors used 5 slides per sample with at least 60% of tumor cells for DNA extraction. How did the authors evaluate the percentage of tumor cells per slide? Also, is there any study evaluating the effect of FFPE on the methylation of MGMT when compared with frozen sections?
9) What were the intervals in fluorescence/scatter measurements during heating? Also, did the authors process the data in any way (e.g. by applying interpolation?) that would enable them to calculate the first derivate? The only data processing mentioned in referenced publication was used to ensure the same temperature alignment for all data.
10) In the previous study (Tsvetkov et al., Cancers 2021; PMID: 33803924) the authors analyzed absorbance at 350, whereas in the current manuscript, they analyzed scatter at 350. What is the rationale behind this change? Is scatter analysis more informative/shows more variability between the samples than absorbance?
11) How exactly were the data processed by AI? Is each type of data (fluorescence at 330, fluorescence at 350, scatter at 350, the fluorescence ratio and their derivates) analyzed separately, or are all included when developing algorithms? Additionally, did the authors ever evaluate the impact of sex or age, on the nanoDSF results, and thus do they include those data when developing the algorithms?
12) The graph presented in Figure 2 should also contain 95% CI or standard deviation to allow the readers to evaluate the data variability among the patients included in the study.
13) Why do the authors only present the data for the first derivate of the F330/F350? What about all the other data? Please include them as supplementary data either in numeric form or at least as similar graphs as presented in Figure 2. This will allow the readers to better understand the direction of changes in the fluorescence/scatter measurements of the plasma samples of glioma patients during heating.
14) What do authors mean by nanoDSF being "non-expansive"? (Page 6, line 206)?
15) How do differences in total plasma protein concentration affect the data obtained by NanoDSF? Did authors measure protein content in analyzed samples?
16) Considering the blood-brain barrier and the low number of circulating tumor cells, what is the authors' hypothesis on the possible mechanism(s) explaining how the aberration in EGFR could affect denaturation profiles?
Author Response
see attachement

Reviewer 2 Report
This work builds on previous report on denaturation plasma profile of glioma patients measured by nanoDSF and development of machine learning algorithm that allows for automatically distinguish glioma patients from healthy controls. Here, the authors show the ability of AI on PDPs to discriminate EGFR alterations predicted with 90% accuracy, but not MGMT methylation status of GBM that in, my opinion, is the main result of the present work suggesting the potential of the used approach to predict GBM.
Points to be considered:
- It is worth giving some details (the meaning of F330 and F350, the ratio F330/F350, although trivial for the authors and some readers) on nanoDSF (in 2.4. Sample Analysis by NanoDSF section), at least for readers who are not familiar with the method.
- I think the authors need to interpret the data presented in Figure 2; what can be the origin of the similarity of post-surgery GBM and aneurysm PDPs, and the difference between the preoperative aneurysm and GBM plasma PDP?
What does actually post-surgical PDP suggest? How the strong increase in the F330/F350 ratio can be explained? Suggestion was made that post-surgery PDP modifications “could be related to neuro-surgery and not specific disease”, however, it should not be the surgery itself, but modifications in plasma related to medication during / post-surgery etc.
- The authors should clarify the choice of PDP measurement 48h after surgery, moreover as pointed out the post-surgery results are not conclusive.
- It is recommended to represent the conclusion based on the results obtained. The conclusion that “the use of PDPs could help the medical decision of physicians and neurosurgeons to propose more aggressive definitive diagnostic and therapeutic approaches” sounds a bit far-fetched, particularly proposing a therapeutic approach having in mind the similar post-surgery PDP of GBM and aneurysm.
- Statistical analysis is lacking.
- All abbreviations should be proceeded by the full names / terms used.
Reviewer 3 Report
Please see the attached PDF. I have copy/pasted the document information in case of conflicts.
Reviewer Comments:
The manuscript is an excellent step forward for the use of ML and AI for thermal liquid biopsy (TLB). The presented problem of glioblastoma diagnosis is a clear area of need that is well presented in the manuscript. The fundamental idea of the paper is to assess how well ML/AI methodologies perform as classifiers for disease differentiation. Although there is novelty in the presentation and application of these methods to thermal liquid biopsy, the methodologies are not presented with enough clarity to justify the conclusions of successful modeling. Improvements to the presentation of the ML/AI work will solidify this work as important to the thermal liquid biopsy community.
Major revisions required within the context of model building and model conclusions. The impact of the work for glioblastoma diagnosis is very high, but the lack of rigor presented within the modeling process does not justify these models were properly constructed and trained and if they will properly adapt to larger data sets.
Major Issues:
There is no presentation of a sensitivity analysis for any statistical, ML, or AI method. Information should be given as to how the models are properly tuned (hyperparameter selection and effects on final outcome). The manuscript continually says the models were trained to a data set but does not give any information regarding rigor behind the training/testing and selection of final models. Please provide a sensitivity analysis for each methodology presented, including how hyperparameters were chosen.
Presented results for LR and SVM methods are not well established for this problem. LR and SVM models are highly dependent on correlation structures within the data. PDPs are notoriously high correlation between temperature readings, even at the first derivative level, yet no accounting of this problem was done when ruling out LR as a useful methodology. No presentation of penalization for dealing with highly correlated variables is discussed. LR and SVM results will likely improve if penalization is incorporated into the modeling process.
RF hyperparameters and discussion of RF tuning is not presented.
AdaBoost is known to be a difficult methodology to properly tune. The authors suggest that AdaBoost is significantly outperforming other methods for EGFR alterations, but this has not been well established outside of the LOOCV accuracies.
No discussion of sample variation is presented. Mean curves demonstrate clear changes to the mean behavior, but PDPs typically have high variation between patients. Please present a discussion of variation to accompany the results shown in Figure 2. Variation is often inflated upon taking derivatives.
Discussion of how derivatives were constructed is not presented. There are multiple methodologies for preparing derivative curves (Functional Data Analysis, Finite Difference Curves). Please include language to clarify how derivatives were achieved.
Minor Issues:
Very small sample sizes are reported with significant imbalance. Consider further discussion on how the imbalance plays a role in the final results (this is only briefly mentioned). Other suggestions here are to use loss-functions that are better suited to imbalance (such as the recently reported focal-loss), or to use resampling methodologies to improve the results around imbalanced data.
Leave-one-out cross-validation is considered an upper bound of error rates. This was understandably used due to the nature and complexity of the data. However, the findings are less trustworthy from LOOCV to ensure these results will properly adapt to larger data sets.
Many of the final models present accuracies that reflect the imbalanced nature of the data set without showing clear signs of differentiation between sample types. Certain presented methods classify nearly all samples into one group (very low true negative, very high true positive). This is indication of improper model selection and tuning (see major issues).
Additional Comments:
The term ‘automatic’ is used through the manuscript but is not clear on what the author’s intend. Does this indicate the entire process from sample to prediction is automated? It seems to this reviewer they are indicating they used the methodologies automatically, without change and using only default settings, which largely detracts from ML/AI success. Please either remove the automatic language or further justify what is meant by automatic.
Remove the language ‘often well-performing’ prior to Support Vector Machines (P3 L116).
The abbreviation GBM is used without introduction.
Small typos and type setting are present in the document. Font changes on P2 L86.
Author Response
see attachement

Round 2
Reviewer 1 Report
The Authors responded to all Reviewer's comments and make appropriate changes to the manuscript.
Author Response
thank you very much
Reviewer 2 Report
The corrections made by the authors impoved the manuscript. I suggests their study can be published.
Author Response
Thank you very much
Reviewer 3 Report
The authors defend their comments with significant rigor in the cover letter. After thorough consideration, I feel the novelty of the PDP results are important for TLB literature, but feel that two key conclusions require additional justification:
1 - the signatures used for AI inputs are more variable than is presented in the manuscript and the conclusion that dedicated signatures have been found requires additional support. Mean signatures have been found, but based on variability, additional statistical modeling is required to validate that these signatures are distinct at this time.
2 - the AI approach is interesting but the model results have large variability and additional rigor to the AI process is required beyond the 'automatic' output from the instrument. There are other methods and schemes that should be considered to improve the AI / classifier conclusions. The low sample limitation that is stated is not limiting in evaluating the AI approach
I suggest additional major revisions of the manuscript with more rigor of the AI approach, inputs, and classification results , even under limited sampling conditions.
The approach presented in the manuscript requires a greater understanding of the data and results (i.e., how derivatives are produced) and more attention to the importance of inputs to AI methods (feature importance and potential feature reduction) along with the impact hyperparameters have on final results. There is no reason that the depth or number of trees couldn't be altered and LOOCV calculated again, over a range of hyperparameter sets, providing a much-needed sensitivity analysis. This would strengthen the critical AI and modeling components of this project.
Below are suggested changes that could solidify this work for publication.
Major Revision 1 – Focus on the PDP signature and perform statistical modeling to statistically conclude that population signatures are distinct.
The authors state that dedicated signatures have been found for the clinical groups examined in the study. Before proceeding with AI and modeling, more rigor should be given to demonstrate this unique finding and ensure a robust understanding of the inputs. The authors at this time do not provide an analysis of the AI inputs and use the results directly from the instrument without knowledge of the technical details involved in producing the derivative curves. For dissemination of data for other research groups to reproduce, it is essential to provide the technical details as to how the instrument data is processed. The authors comment that the manufacturer of the nanoDSF machine will not provide the details of the computation of the derivative of the different signals is disappointing, but the authors provide a path forward in having obtained similar results using the finite difference method. Here are suggestions for solidifying the conclusion of distinct PDP signatures. Assess the resulting curves from different populations critically using statistical testing. Point-wise testing with correction could be used to ascertain which elements of the curve are distinct between populations. This could help reduce the input size to the AI and improve results, specifically with traditional modeling such as LR. The curves could also be assessed as functional objects using Functional Data Analysis. Tests are readily available from FDA that can ask if the curves are distinct (known as a Functional ANOVA). Based on supplemental material and the variance of the curves displayed, it is unlikely that functional-ANOVA would find distinct curves, but this is future work that can be considered by the authors. This will again challenge the authors to evaluate truncated PDPs as improved inputs for AI. More work should be placed into describing the characteristics and differences between the curves for each clinical group, which can guide updates and improvements for AI inputs.
Major Revision 2 – Improve model validation and sensitivity even under limited sampling conditions.
The authors use AI methods, specially ADABoost, to try and draw the conclusion that the AI will work well for class identification. Authors are correct to use LOOCV, but reluctant to evaluate how hyperparameters effect the results. It would not be too burdensome to prepare a set of hyperparameters to evaluate, over a large grid, and calculate LOOCV rates for each hyperparameter set. This can help obtain 1) how hyperparameters influence the output and predictions and 2) how generalizable these results are for future larger sampled results. Hyperparameters have been chosen based on earlier publications, which is a good starting point. However, if small changes to hyperparameters cause significant loss in classification performance, this is a sign of overfitting and improper input design. The models are likely to be sensitive to hyperparameters, but by assessing many alternative hyperparameter sets, will improve their knowledge of how models perform over a range of parameters and solidify the conclusion that these models will generalize with improved sampling.
Some supplementary material could be prepared to present the results of the sensitivity analysis based on LOOCV, but it would suffice to include details in the manuscript to describe what is found here. Sensitivity analysis should be conducted for SVMs, RFs, and ADABoost methods.
Major Revision 3 – Feature importance for improved AI input understanding.
After finalizing ADABoost and RF models, potentially with improved inputs, the authors should provide an evaluation of feature importance. The curves provided in supplemental material have very high variability in several regions, with only minimal regions of distinction. Feature importance will allow the authors to provide a focus of which parts of the PDPs are critical for distinction. The authors must ensure that the key features relate to important temperature regions that are distinct between populations. Feature importance may also find that more noisy regions with significant overlap are important, which is again an indication of overfitting, and that the boosting (or tree-based modeling) are finding answers from within the noise, which will detract from the concept that these models can generalize. If truncated input regions are used, such as focusing on the regions of the PDPs that are distinct between populations, then one can be assured that the AI is using inputs that are key for differentiation, and not fitting noise in this low sample investigation.
I believe these three major revisions will help solidify the two major conclusions discussed by this reviewer, along with appropriate discussion in the manuscript of the suggested work and the strengths and limitations of the study.
Author Response
We thank the reviewer for this new round of reviews to which we gave a detailed answer in the attached file.

Round 3
Reviewer 3 Report
Although there are several areas that have been requested for improvement, the authors have provided a suitable rebuttal for the revisions and I have no further constructive comments at this time. I strongly encourage the authors review what was asked and implement these ideas in future work. The authors should be very cautious moving forward with their AI as they have not clearly shown a balance in Bias-Variance and are likely finding very high variance models that will require significant rigor for extending to larger data sets properly. This is a major reason AI systems fail in implementation, yet the authors have not chosen to produce the important downstream steps of AI development and incorrectly claim the method suggested is unethical. They are instead unclear as to what was asked for by this reviewer and failed to produce any results requested. Newly provided details of their AI implementation continues to indicate they are not properly fitting or evaluating any of the model types presented. Reviewer concerns regarding final model estimates were again disregarded for small sample size arguments. There is very little rigor behind the AI.
However, they have provided a unique and important new research goal for thermal liquid biopsy. I continue to have worries regarding their AI descriptions and execution, but find their goals of a preliminary study to be adequate at this time.